# A Framework to Predict Gastric Cancer Based on Tongue Features and Deep Learning

**DOI:** 10.3390/mi14010053

**Published:** 2022-12-25

**Authors:** Xiaolong Zhu, Yuhang Ma, Dong Guo, Jiuzhang Men, Chenyang Xue, Xiyuan Cao, Zhidong Zhang

**Affiliations:** 1Key Laboratory of Instrumentation Science & Dynamic Measurement, School of Instrument and Electronics, North University of China, Taiyuan 030051, China; 2Shanxi University of Chinese Medicine, Taiyuan 030051, China

**Keywords:** gastric cancer, tongue features, non-invasive, prediction framework, deep learning

## Abstract

Gastric cancer has become a global health issue, severely disrupting daily life. Early detection in gastric cancer patients and immediate treatment contribute significantly to the protection of human health. However, routine gastric cancer examinations carry the risk of complications and are time-consuming. We proposed a framework to predict gastric cancer non-invasively and conveniently. A total of 703 tongue images were acquired using a bespoke tongue image capture instrument, then a dataset containing subjects with and without gastric cancer was created. As the images acquired by this instrument contain non-tongue areas, the Deeplabv3+ network was applied for tongue segmentation to reduce the interference in feature extraction. Nine tongue features were extracted, relationships between tongue features and gastric cancer were explored by using statistical methods and deep learning, finally a prediction framework for gastric cancer was designed. The experimental results showed that the proposed framework had a strong detection ability, with an accuracy of 93.6%. The gastric cancer prediction framework created by combining statistical methods and deep learning proposes a scheme for exploring the relationships between gastric cancer and tongue features. This framework contributes to the effective early diagnosis of patients with gastric cancer.

## 1. Introduction

Gastric cancer is one of the most prevalent malignancies in humans, and the reason for many deaths each year [1]. Early-stage gastric cancer patients are asymptomatic [2], and a few present non-specific symptoms, such as epigastric discomfort and belching. The symptoms tend to be ignored because of their similarity to symptoms of chronic gastric disease. Therefore, identifying gastric cancer patients early, providing immediate treatment, and delaying the deterioration of gastric cancer patients to advanced stages are crucial in the protection of public health.

The most commonly used method for the detection of gastric cancer is gastroscopy [3]. By observing the condition of the gastric mucosa, the physicians assess the severity and location of the lesion. However, extensive expertise and experience are required from physicians to identify suspicious lesions, and the diagnosis results may be affected by the working state of physicians (e.g., alertness) [4]. The advancement of modern technology has enabled researchers to apply artificial intelligence techniques to gastroscopy [5,6]. Compared to specialized endoscopists, artificial intelligence technology is faster in assessing the severity of the lesions and more accurate. However, gastroscopy is an invasive operation that can potentially cause complications, such as perforation and bleeding [7]. Patients with minor symptoms are reluctant to choose gastroscopy to detect gastric cancer.

Tongue diagnosis is a non-invasive diagnostic modality to diagnose the status of patients conveniently. Tongue features not only reveal the physical condition of the organs but also correlate with their functions [8]. Significant changes in tongue features are observed when people develop physical lesions. Practitioners assess the severity of lesions and formulate solutions for the corresponding disease by analyzing the changes in the tongue features of patient [9].

Recently, instead of observing and recording tongue features, research has been focusing on objective tongue feature analysis with the benefit of modern technology. Tongue features have been digitally analyzed to discover convenient ways to detect diseases [10,11,12]. The extensive application of artificial intelligence techniques [13] in the medical field has increased objectivity in the interpretation of the relationship between tongue features and diseases. Image processing [14] has been utilized to extract information from tongue images and capture physical features that cannot be detected by human eyes. Deep learning [15] has been applied to automatically identify and learn tongue features. Links have been established between tongue features and diseases based on these technologies to predict diseases, such as breast cancer [16], diabetes [17], and gastric cancer [18].

Studies exploring the links between gastric cancer and tongue features have contributed to the detection of gastric cancer. Gastric cancer has been confirmed to be correlated with microbiota in the oral cavity, such as helicobacter pylori [19,20]. The microbiota in the oral cavity of gastric cancer patients has been analyzed by utilizing high-throughput sequencing and the tongue coating thickness has been measured in the literature [21,22]. As such, an association between tongue features and microbiota was established for gastric cancer detection. However, the sensitivity of this method is low and tends to be susceptible to microbial density. Gholami et al. [23] proposed a method that combined artificial intelligence techniques and tongue features to detect gastric cancer. The proposed method had a high accuracy in distinguishing between patients and healthy subjects. Nevertheless, the adaptive reference model that was utilized might change the tongue features, e.g., tongue shape and tongue color. 

The main contribution and innovation of this study is to investigate the feasibility of predicting gastric cancer by the tongue. Because no open-source dataset of gastric cancer tongue images exists, a tongue image capture instrument was used to acquire tongue images of gastric cancer subjects and non-gastric cancer subjects. Based on image pre-processing and image enhancement, an original dataset was constructed. Next, the image segmentation algorithm was used for the tongue segmentation task. Then, nine tongue features were extracted with guidance from professional physicians. The association between these tongue features and gastric cancer is explored by using differential analysis, importance analysis and correlation analysis. Four tongue features that contribute to the prediction of gastric cancer are screened and retained. These four features including tongue shape, saliva, tongue coating thickness and tongue coating texture are used as a basis for predicting gastric cancer. Finally, a framework using EfficientNet [24] is established to achieve non-invasive prediction of gastric cancer patients. The details of the entire framework are shown in Figure 1. It provides an approach to study the relationship between tongue features and gastric cancer.

## 2. Related Works

Gastric cancer is an enormous threat to human health. Early detection of gastric cancer has a positive significance in reducing the mortality rate of gastric cancer patients. 

As one of the effective methods to diagnose gastric cancer, the physician can visually observe the changes of gastric mucosa and the size of the lesions. However, the detection results are affected by the work experience and subjective perception of physicians. To overcome these limitations, researchers [25,26,27] built gastric cancer identification models to detect gastric cancer in endoscopic images by using artificial intelligence techniques. The model was trained based on a large number of endoscopic images of gastric cancer annotated by professional physicians. The experimental results show that the model can accurately identify a large number of gastroscopic images within a short time. However, gastroscopy is an invasive operation that can easily lead to complications. The detection and analysis of volatile organic compounds in exhaled breath is a non-invasive method for diagnosing gastric cancer. Huang et al. [28] used a tubular surface-enhanced Raman scattering sensor to capture volatile organic compounds in human exhaled breath to noninvasively screen patients for gastric cancer, with an accuracy of 89.83%. The results showed that the breath analysis method provides an excellent option for screening gastric cancer. However, the results of breath analysis are susceptible to factors such as the breathing collection method, patient physiological condition, testing environment and analysis method [29].

Jiang et al. [30] constructed a noninvasive, high-precision diagnostic model for gastric cancer, using characteristics such as age, gender, and individual behavioral lifestyle. Compared to other models, the model they built using an extreme gradient-based augmentation algorithm obtained better results. The model achieved an overall accuracy of 85.7% in the test set. Zhu et al. [31] also used machine learning to build a noninvasive prediction model for gastric cancer risk. They used statistical methods to screen for significant features and performed multivariate analysis of significant features to exclude features that were not useful for predicting gastric cancer. The input features of the above models were usually the basic physiological information of the subjects. However, using tongue images to build disease prediction models has been neglected [11,32,33]. As a basic organ of the human body, changes in tongue features reflect physiological and case information. The diagnosis of tongue features can be used as a non-invasive screening method for the detection of gastric cancer. Gholami et al. [23] used tongue color and tongue lint to build a prediction model for gastric cancer, which had an accuracy of 91%. However, they only studied the association between tongue color and tongue lint with gastric cancer without exploring the association between other tongue features and gastric cancer. 

Therefore, we extracted nine tongue features and explored the relationship between them and gastric cancer. Through differential analysis, importance analysis, and correlation analysis, tongue shape, saliva, tongue coating thickness and tongue coating texture were finally selected as the input features for predicting gastric cancer. In addition, we used image segmentation to remove non-tongue regions before building the gastric cancer prediction framework to reduce interference and improve the efficiency and accuracy of prediction.

## 3. Methods

To ensure the established framework had clinical application, we used an instrument to collect the tongue images of gastric cancer patients and non-gastric cancer subjects in hospitals. The images captured by the tongue image acquisition instrument contain non-tongue areas such as the face and the instrument. Non-tongue regions are significantly reduced by image pre-processing, which is helpful to improve the efficiency of image segmentation. The processed tongue images were input into the gastric cancer prediction framework, then the final prediction results were obtained.

### 3.1. Date Sources

Data was collected from Shanxi Cancer Hospital and the Affiliated Hospital of Shanxi University of Traditional Chinese Medicine from January 2021 to August 2022, with 703 images in total. The dataset consisted of 103 tongue images of gastric cancer patients and 600 tongue images of non-gastric cancer subjects that included patients with other diseases and normal individuals. The diseases of these subjects were definitely diagnosed. Tongue images of the patient were captured under the guidance of specialized physicians. The subjects were arranged to sit in front of the instrument and extend their tongue naturally (Figure 2). During the capture process, the quality of the tongue images was strictly controlled. The light environment was kept stable during image acquisition and blurred images were removed to ensure high-quality tongue acquisition. After image acquisition from each subject, the instrument was disinfected and ventilated to ensure a hygienic and safe acquisition environment.

### 3.2. Date Preprocess

The resolution of the acquired tongue image was 3264 × 2488 with horizontal and vertical resolution of 96 dpi. High-resolution tongue images containing non-tongue regions consume a lot of computational resources in the deep learning network. The tongue region extraction method proposed by Li [34] was employed to greatly reduce the area of non-tongue regions. This method improved the efficiency of subsequent tongue segmentation, while reducing the computational load of the deep learning network. After cropping the original tongue images, the image annotation software was used to construct a tongue dataset for segmentation model training. To avoid overfitting of the prediction model due to data imbalance, the number of tongue images was expanded to 4375 utilizing data augmentation [35]. The tongue images of the subjects with gastric cancer and non-gastric cancer were expanded in the same ratios, and the image augmentation algorithms used in this study include geometric transformations (e.g., flipping and panning) and the addition of Gaussian noise. The image augmentation algorithm is implemented using the imgaug library in python.

### 3.3. Segmentation of Tongue Images

In addition to the tongue region, images acquired by the tongue image capture instrument frequently contained areas, such as lips and teeth. The tongue-part of the images was segmented to eliminate the influence of non-tongue areas on the subsequent analysis. Existing automatic tongue segmentation approaches are classified into two categories, namely segmentation models based on traditional methods, such as the region growing method and the thresholding method, and segmentation networks based on deep learning. Traditional segmentation methods are insensitive to the color of the regions close to the tongue and easily mistake these regions for the tongue [36,37,38]. In this stage, Deeplabv3+ [39] network, which is based on deep learning, was chosen for the tongue segmentation task, and clinically collected tongue images were used as the dataset for segmentation.

The Deeplabv3+ network is based on an encoder-decoder architecture. The encoder part consists of a network for extracting features and multiple parallelly dilated convolution layers. The MobileNetV2 [40] is used as the backbone network. In the encoder part, tongue image features are extracted to generate high-level semantic features. To enhance the learning ability of the network, the low-level semantic features are merged with the high-level semantic. The merged feature map is then processed by the convolution and up-sampling layers to form the last semantic segmentation map. The architecture of the Deeplabv3+ network is shown in Figure 3. 

The data set used for tongue segmentation was divided in a 7:2:1 ratio into the training set, validation set and test set. The MobileNetV2 network weight trained on ImageNet was used as the beginning weight. The Adam optimizer [41] was used to optimize the model during the training process. The segmentation network was trained with 200 epochs. In the first 100 training epochs, the weights of the MobileNetV2 network were frozen with a learning rate of 1 × 10^−2^. In the last 100 epochs of unfrozen training, the learning rate of the global network was 1 × 10^−4^.

### 3.4. Extraction and Analysis of Tongue Features

Combining the professional opinions of several doctors, nine tongue features were extracted to explore the difference between the tongues of gastric cancer patients and non-gastric cancer subjects. The extracted tongue features are listed in the following paragraphs.
Tongue shape: fat tongue, thin tongue and normal tongue;Tooth-marked tongue: tooth mark and normal;Spots and prickles tongue: Spots and prickles and normal;Saliva: dry and normal;Tongue coating thickness: thick and thin;Tongue coating texture: greasy and normal;Tongue color: pale white, pale red, red and deep red;Tongue coating color: white and yellow;Tongue Fissure: fissured tongue and normal.

The differences in tongue features between the gastric cancer patients and the control group were contrasted using statistical methods. The cross-plot of tongue features between the two groups is shown in Figure 4. Saliva, tongue coating thickness, tongue fissure, tongue coating texture, and tooth-marked tongue were distinctly different between the two groups. The weight of these tongue features in patients with gastric cancer varied considerably from that in the control group.

The tongue features of the two groups were contrasted using chi-square test and independent t-tests to explore features that were significantly different (*p*-value < 0.001). Statistics were based on the *p*-value obtained by the significance test method, with *p*-value < 0.001 considered as a highly significant difference, 0.001 < *p*-value < 0.005 considered as a significant difference, and *p*-value > 0.005 considered as no significant difference.

The differences in tongue features between patients with gastric cancer and the control group are shown in Table 1. Among them, five tongue features, namely tooth-marked tongue, saliva, tongue coating thickness, tongue coating texture, and tongue shape, were significantly different between the gastric cancer patients and the control group (*p*-value < 0.001). The proportion of yellow tongue coating in gastric cancer patients was significantly higher than in the control group (51.46% vs. 35.83%, 0.001 < *p*-value < 0.005). In addition, the proportion of spots and prickles tongue in gastric cancer patients was significantly higher than in the control group (36.89% vs. 24.17%, 0.001 < *p*-value < 0.005). No significant difference was detected between patients with gastric cancer and the control group in tongue color and tongue fissure (*p*-value > 0.005).

The XGBT algorithm [42] was utilized to calculate the importance of each tongue feature. Feature importance measures the contribution of each input feature to the prediction result of the model and can highlight the relevance of the feature to the target. Features with lower importance scores were removed, while those with higher importance scores were retained. Filtering tongue features that had an impact on the prediction with the feature importance analysis method reduced the quantity of features input into the neural network model, which contributed to the computational efficiency and accuracy of the model. Figure 5 shows the importance of the tongue features results. The importance scores of features such as dry saliva, fat tongue, and white tongue were high and have a major impact on the model prediction results. The importance scores of pale white tongue, deep red tongue, yellow tongue coating, tooth mark tongue, and spots and prickles tongue were low and are not shown in the figure.

Subsequently, the association between the nine tongue features of the gastric cancer patients was explored. Correlation analysis was applied to analyze two or more features that were correlated to measure the closeness of the correlation. The correlations between the tongue features of gastric cancer patients are shown in Figure 6. Darker colors in the correlation graph represent a higher correlation between two tongue features. Results show that there is a high probability of association between the thick tongue coating and the greasy tongue coating, while the remaining tongue features were not significantly associated with each other. A negative value for the correlation between two features indicates that an increase in one tongue feature causes a decrease in the other feature.

Utilizing statistical methods and artificial intelligence techniques for tongue feature analysis, the tongue features with significant differences, high importance scores and high correlations between the gastric cancer patients and the control group were selected as input features for the gastric cancer prediction framework. Finally, tongue shape, saliva, tongue coating thickness and tongue coating texture were also used as features to build a deep learning gastric cancer prediction framework.

To avoid the shortcomings of single feature scale, easy saturation of dimensional scaling and poor classification effect in traditional image classification algorithms, the EfficientNet network was used to classify gastric cancer and non-gastric cancer tongue images. Compared with different CNN models, the EfficientNet model successfully achieved a higher accuracy and efficiency. It used a simple and efficient composite coefficient to scale the network in three dimensions: network depth, network width and image resolution. The MBConv block in MobileNetV2 was used as the backbone of the EfficientNet network. The MBConv block was composed of two convolutional layers, a depth-separable convolutional layer, and a feature extraction module. The basic network architecture of EfficientNet-B0 is shown in Figure 7. Meanwhile, in order to avoid the problem of overfitting and instability due to an insufficient number of image samples, batch normalization and dropout were used to reduce the dependency between convolutional layers, reduce the activity of some neurons in the training process, and suppress the occurrence of overfitting, thus improving the generalization ability of the network and enhancing the robustness of the model classification.

The dataset used to build the prediction framework was divided in an 8:1:1 ratio into the training set, validation set and test set. When the EfficientNet network model was trained on the dataset, the official weight was used as the initial weight of the model. After 200 epochs, the training ended with a learning rate of 1 × 10^−4^. At the end of the framework, the Softmax classifier normalized the two classification results and the output values were transformed into probabilities. In the final output layer, the output value with the highest probability was selected as the predicted result.

### 3.5. Performance Metrics

The experiments were performed on a Windows 10 operating system, Intel Core i7-10700, NVIDIA GeForce RTX 2060, based on Pytorch deep learning framework.

In this section, the effectiveness of the segmentation method and gastric cancer prediction framework was assessed with four metrics. The performance of the Deeplabv3+ network was evaluated using the mean intersection over union (MIou) and mean pixel accuracy (MPA) [43]. Accuracy and F1-score were used to evaluate the classification effectiveness of the gastric cancer prediction framework. The definitions and calculations for MIoU, MPA, Accuracy and F1-score are as follows.

MIou is the standard evaluation method for segmentation methods. The specific value of the intersection of the predicted segmentation and ground truth of each class of pixels to the union set is calculated. The ratios of all classes are then summed and averaged to obtain MIou:(1)$$\begin{array}{l} \mathrm{MIoU}=\frac{1}{k+1}\sum_{i=0}^{k}\frac{p_{ii}}{\sum_{j=0}^{n}p_{ij}+\sum_{j=0}^{n}p_{ji}-p_{ii}} \end{array}$$
where *k* is the number of categories in the image except the background, *p_ii_* is the number of pixels belonging to category *i* that are predicted to be in category *i*, *p_ij_* is the number of pixels belonging to category *i* that are predicted to be in category *j*.

MPA is based on pixel accuracy (PA). The specific value of the number of pixels correctly predicted in each class to the total number of pixels was calculated to obtain the PA. The PAs of all classes were then summed and averaged to obtain the MPA:(2)$$\mathrm{MPA}=\frac{1}{k+1}\sum_{i=0}^{k}\frac{p_{ii}}{\sum_{j=0}^{n}p_{ij}}$$

Accuracy is the proportion of samples correctly predicted by the model to the total sample:(3)$$Accuracy=\frac{TP+TN}{TP+TN+FP+FN}$$
where TP is the number of positive samples that are correctly identified, TN is the number of negative samples correctly identified, FP is the number of negative samples that are false positives and FN is the number of positive samples that were identified as negatives.

The F1-score is a performance evaluation index of a binary classification model. It is a balance between the precision and recall giving a more balanced metric by calculating the harmonic mean of precision and recall:(4)$$\mathrm{Pr}ecision=\frac{TP}{TP+FP}$$
(5)Recall=TPTP+FN
(6)F1−score=2×Precision×RecallPrecision+Recall

## 4. Results

To validate the segmentation effect of the Deeplabv3+ network, other models commonly used for medical image segmentation were built for comparison with the Deeplabv3+ network. The segmentation networks were evaluated with MIou and MPA. The segmentation effect of the three networks evaluated is shown in Table 2. The Deeplabv3+ network made comprehensive use of the MobileNetV2 network and atrous convolution, which minimizes the information loss in the process of image segmentation. The Deeplabv3+ network outperformed PSPNet and U-Net in terms of Mlou and MPA, showing that the Deeplabv3+ network model has superior segmentation ability.

The validity of the gastric cancer prediction framework was tested on a validation set containing tongue images of gastric cancer patients and tongue images without non-gastric cancer. Seventy tongue images in the dataset were used to test the effectiveness of the framework. Representative samples of the tongue images in the test set are shown in Figure 8.

The tongue images of gastric cancer patients and the control group in the validation set were input into the framework to assess its prediction effect. The accuracy of the framework was 93.6%, and the F1-score was 92.6%. Table 3 shows the prediction results of the four tongue images in Figure 8. The prediction framework has high accuracy in identifying tongue images of gastric cancer and tongue images of the control group.

## 5. Discussion

A gastric cancer prediction framework was built by statistical analysis of tongue features and deep learning. The framework has high accuracy in detecting gastric cancer patients and the detection process has the advantage of being non-invasive. This study presents a novel approach to exploring the relationship between gastric cancer and tongue features, which has clinical application.

The presented framework has a strong detection ability, but two aspects can be continuously improved. A major part of the time was spent collecting the images in hospitals, but the number of tongue images collected from gastric cancer patients was still low, which is a common problem in the area of medical image studies. Second, tongue images were collected using a standard capture instrument, but the tongue extension posture of each patient has an impact on subsequent tongue feature extraction.

## 6. Conclusions

Cancer is a major public health problem worldwide. The penetration of artificial intelligence technology into the medical field has become more convenient and accurate to detect gastric cancer. In this paper, we proposed a prediction framework for gastric cancer based on tongue features and deep learning. The framework predicts gastric cancer patients non-invasively and conveniently. Tongue images are acquired by using a standard tongue image capture instrument, and the Deeplabv3+ network is applied to accurately segment the tongue region. Nine tongue features are extracted with guidance from professional doctors, and the relationships between tongue features and gastric cancer are explored by using statistical methods and deep learning. Through differential analysis, importance analysis, and correlation analysis, tongue shape, saliva, tongue coating thickness and tongue coating texture were finally selected as the input features for predicting gastric cancer. Finally, the EfficientNet network was used to build a gastric cancer prediction framework. The experimental results showed that this framework could accurately distinguish between gastric cancer patients and non-gastric cancer subjects with an accuracy of 93.6%. Compared to other similar work, our framework incorporates more tongue features and can more comprehensively predict gastric cancer.

Future work will focus on acquiring more tongue images of gastric cancer patients to extract and further analyze tongue features. Meanwhile, we should improve the framework structure and enhance the robustness of the framework to obtain more accurate gastric cancer prediction results.

## Figures and Tables

**Figure 1 micromachines-14-00053-f001:**
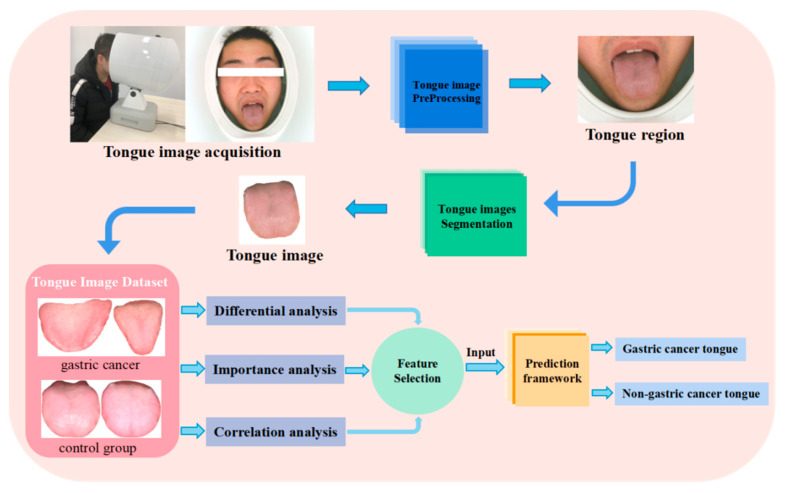
Overview of our framework.

**Figure 2 micromachines-14-00053-f002:**
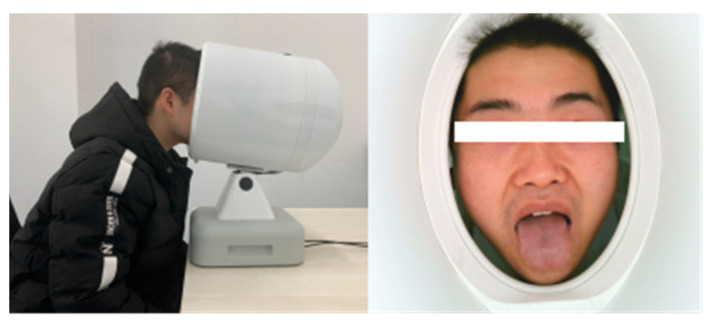
The process of capturing tongue images. During the collection, the subjects place their mandibles on the instrument and extended their tongues naturally.

**Figure 3 micromachines-14-00053-f003:**
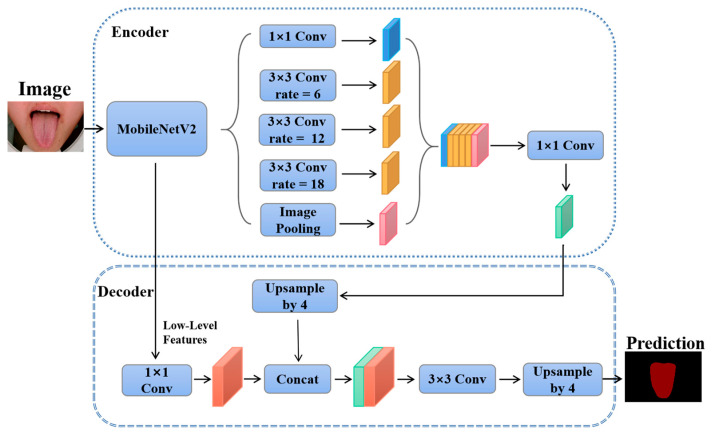
Structure of Deeplabv3+. The high-level semantic features and the low-level semantic features of the tongue image are extracted and fused by the Deeplabv3+ network. The low-level features have high resolution and contain a wealth of detailed information about the image. The high-level features have stronger semantic information. The fusion of features of different scales is an important method to improve the performance of segmentation networks.

**Figure 4 micromachines-14-00053-f004:**
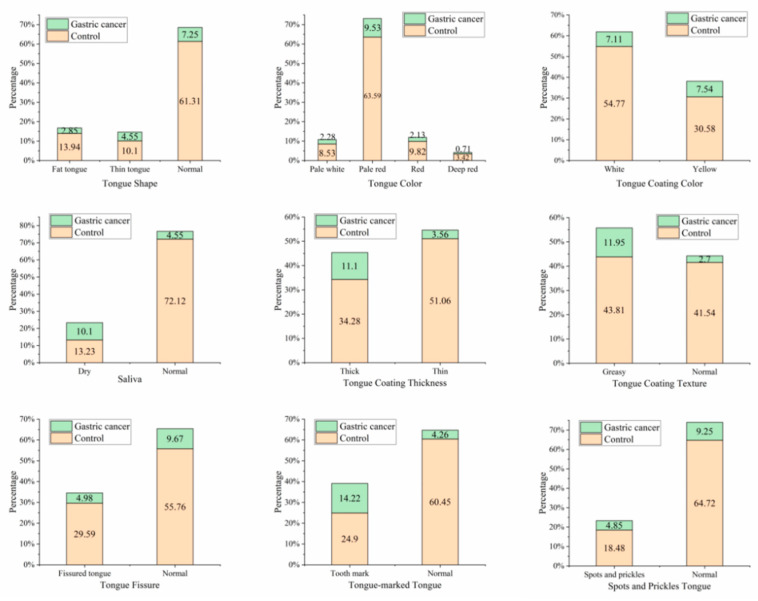
The cross-plot of tongue features between gastric cancer group and non-gastric cancer group. In the cross-plot, the green section represents gastric cancer patients and the orange section represents non-gastric cancer subjects. In the vertical direction, the proportions of different tongue features are clearly shown. In the horizontal direction, the distribution of tongue features in the gastric cancer group and the control group are shown.

**Figure 5 micromachines-14-00053-f005:**
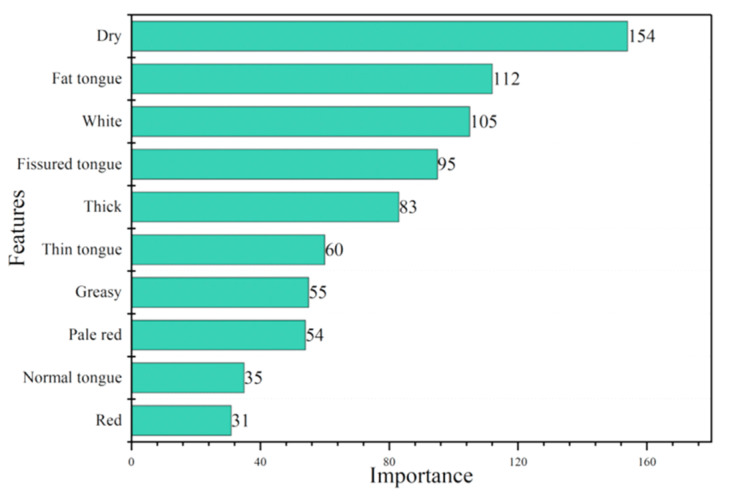
Importance of tongue features.

**Figure 6 micromachines-14-00053-f006:**
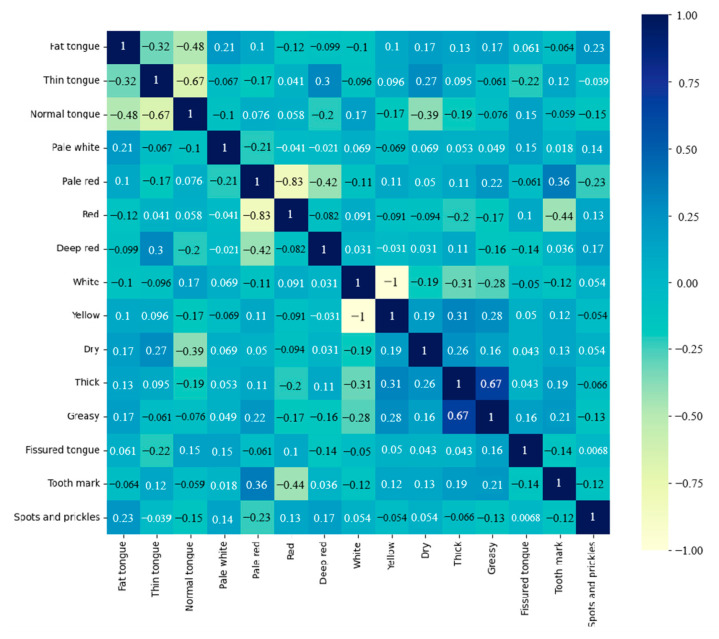
Correlation between tongue features in the gastric cancer group.

**Figure 7 micromachines-14-00053-f007:**
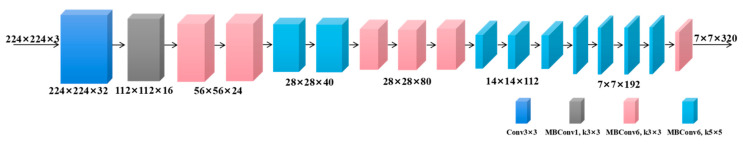
Baseline network for EfficientNet-B0.

**Figure 8 micromachines-14-00053-f008:**
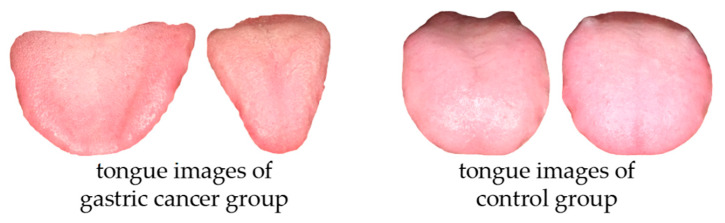
Tongue images in the test set, Image 1 and Image 2 belong in the tongue images of gastric cancer group; Image 3 and Image 4 belong in the tongue images of control group.

**Table 1 micromachines-14-00053-t001:** Comparison of tongue features between patients with gastric cancer and control subjects.

Tongue Features (n, %)		Control n = 600	Gastric Cancer n = 103	Number of Features	*p*-Value
Tongue Shape	Fat tongue	98 (83.05%)	20 (16.95%)	118 (16.79%)	<0.001
	Thin tongue	71 (68.93%)	32 (31.07%)	103 (14.65%)	
	Normal tongue	431 (89.42%)	51 (10.58%)	482 (68.56%)	
Tongue Color	Pale white	60 (78.95%)	16 (21.05%)	76 (10.81%)	0.22
	Pale red	447 (86.96%)	67 (13.04%)	514 (73.12%)	
	Red	69 (82.14%)	15 (17.86%)	84 (11.95%)	
	Deep red	24 (82.76%)	5 (17.24%)	29 (4.13%)	
Tongue Coating Color	White	385 (88.51%)	50 (11.49%)	435 (61.88%)	0.0036
	Yellow	215 (80.22%)	53 (19.78%)	268 (38.12%)	
Saliva	Dry	93 (56.71%)	71 (43.29%)	164 (23.33%)	<0.001
	Normal	507 (94.06%)	32 (5.94%)	539 (76.67%)	
Tongue Coating Thickness	Thick	241 (75.55%)	78 (24.45%)	319 (45.38%)	<0.001
	Thin	359 (93.49%)	25 (6.51%)	384 (54.62%)	
Tongue Coating Texture	Greasy	308 (78.57%)	84 (21.43%)	392 (55.76%)	<0.001
	Normal	292 (93.89%)	19 (6.11%)	311 (44.24%)	
Tongue Fissure	Fissured tongue	208 (85.60%)	35 (14.40%)	243 (34.57%)	0.98
	Normal	392 (85.22%)	68 (14.78%)	460 (65.43%)	
Tooth-marked Tongue	Tooth mark	175 (63.64%)	100 (36.36%)	275 (39.12%)	<0.001
	Normal	425 (99.30%)	3 (0.70%)	428 (60.88%)	
Spots and Prickles Tongue	Spots and prickles	145 (79.23%)	38 (20.77%)	183 (23.33%)	0.0096
	Normal	455 (87.50%)	65 (12.50%)	520 (73.97%)	

**Table 2 micromachines-14-00053-t002:** Results of different segmentation networks.

Method	MPA	MIoU
PSPNet	97.58%	96.6%
U-Net	98.58%	97.14%
Deeplabv3+	98.93%	97.96%

**Table 3 micromachines-14-00053-t003:** The final results of the classification.

Result	Image 1	Image 2	Image 3	Image 4
control	16.9%	13.3%	98.6%	99.2%
gastric cancer	83.1%	86.7%	0.14%	0.08%

## Data Availability

The data presented in this study are available on request from the corresponding author. The data are not publicly available due to this data being supplied by Key Laboratory of Instrumentation Science & Dynamic Measurement (North University of China), Ministry of Education and so cannot be made freely available.

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
