# Peer review of "A Framework to Predict Gastric Cancer Based on Tongue Features and Deep Learning"

_micromachines, 2022, doi:10.3390/mi14010053_

Round 1

Reviewer 1 Report

The paper present an AI framework to detect gastric cancer using subject's tongue images. The paper is well written and clear and the reader enjoys a smooth ride all over the paper. There are only some points that if get addressed, I believe it give a better ride to the readers. 

1. In the first part (introduction) you introduced several works related and then in line 70 of page 2 you introduce your paper. So the question is why your research is needed? 

2. In line 112 if page 3, data expansion is proposed to solve the data imbalanced issue. But there is no reference of such work. Could you discuss it more or provide a reference of such work? How you did it? What do you mean by expansion? Did you use any package for it?

3. References are needed for some specific topics such as MobileNetV2. 

Once revised, I would be happy to read it again and provide my feedback.

Thanks

Reviewer 2 Report

This paper is proposing a framework to predict gastric cancer based on tongue features and deep learning.  the paper is well presented and discussed, however there is some drawbacks need to be improved in the revised version.

1-   Several syntaxes and grammatical need to check, moreover, some sentences are not easy to understand to the reader. Seen line 230-228 page 8. "The Efficient Net network model was trained 229 on the dataset using the default weights as the beginning weight and the training ended 230 with a learning rate of 1 × 10-4, 200 epochs later"

2-   The caption of figure 1 is very long, it is advised to move the description to the text and just make the caption to illustrate the figure.

3-  Don't add heading over heading. Add a few lines related to the detail of a particular section before starting a sub-section. i.e. Don’t add a new section and subsection directly, see section 2 and 2.1.

4-  “The Adam optimizer”,” XGBT algorithm “ the author(s) need to cite the references.

5-   The paper needs a few more related works and could be reviewed to show the significance of the proposed work.

6-  It is not clear how the authors calculated the mean intersection over union (MIou) and mean pixel accuracy (MPA) and in what metric they calculate the accuracy of the proposed system.

7-  More description of the proposed framework is advised to make it clear for the readers. It is encouraged to include more specifications of the framework.

8-  The obtained results need more description and analysis, why only using 4 images in evaluation of the system.

9-  Abstract and conclusions need to improve to illustrate the obtained results and contributions of the work.

Reviewer 3 Report

This research article can consider adding a recent chapter called literature survey and include a few recent works published as part of the literature survey.

Technical aspects in this work are not discussed much

What is the novelty involved in this work?

Better add a novel and key contributions of this work in the introduction part.

Present the proposed algorithm in the methods section

Key works that are going to be included in the literature review part should be also considered for the comparison of the results.

How this proposed methodology will be better compared to other similar works should be a point to be proved here.

Overall significant improvements are required to strengthen this work on the technical front.

Round 2

Reviewer 3 Report

This work has improved significantly over this revision and it can be accepted now.